# Screening of Active Substances Regulating Alzheimer’s Disease in Ginger and Visualization of the Effectiveness on 6-Gingerol Pathway Targets

**DOI:** 10.3390/foods13040612

**Published:** 2024-02-18

**Authors:** Yecan Pan, Zishu Li, Xiaoyu Zhao, Yang Du, Lin Zhang, Yushun Lu, Ling Yang, Yilin Cao, Jing Qiu, Yongzhong Qian

**Affiliations:** 1Key Laboratory of Agro-Product Quality and Safety, Institute of Quality Standards and Testing Technology for Agro-Products, Chinese Academy of Agricultural Sciences, Beijing 100081, China; panyecan.love@163.com (Y.P.); zishuuu3210@163.com (Z.L.); zhaoxiaoyuayu@163.com (X.Z.); 17667199010@163.com (Y.D.); lindeliner@163.com (L.Z.); luyushun@caas.cn (Y.L.); 15085017125@163.com (L.Y.); hmcccy0@163.com (Y.C.); qiujing@caas.cn (J.Q.); 2Key Laboratory of Agri-Food Quality and Safety, Ministry of Agriculture and Rural Affairs, Beijing 100081, China

**Keywords:** 6-Gingerol, ginger, Alzheimer’s disease (AD), network pharmacology, molecular dynamics (MD)

## Abstract

Ginger has been reported to potentially treat Alzheimer’s disease (AD), but the specific compounds responsible for this biological function and their mechanisms are still unknown. In this study, a combination of network pharmacology, molecular docking, and dynamic simulation technology was used to screen active substances that regulate AD and explore their mechanisms. The TCMSP, GeneCards, OMIM, and DisGeNET databases were utilized to obtain 95 cross-targets related to ginger’s active ingredients and AD as key targets. A functional enrichment analysis revealed that the pathways in which ginger’s active substances may be involved in regulating AD include response to exogenous stimuli, response to oxidative stress, response to toxic substances, and lipid metabolism, among others. Furthermore, a drug-active ingredient–key target interaction network diagram was constructed, highlighting that 6-Gingerol is associated with 16 key targets. Additionally, a protein–protein interaction (PPI) network was mapped for the key targets, and *HUB* genes (*ALB*, *ACTB*, *GAPDH*, *CASP3*, and *CAT*) were identified. Based on the results of network pharmacology and cell experiments, 6-Gingerol was selected as the active ingredient for further investigation. Molecular docking was performed between 6-Gingerol and its 16 key targets, and the top three proteins with the strongest binding affinities (*ACHE*, *MMP2*, and *PTGS2*) were chosen for molecular dynamics analysis together with the *CASP3* protein as the HUB gene. The findings indicate that 6-Gingerol exhibits strong binding ability to these disease targets, suggesting its potential role in regulating AD at the molecular level, as well as in abnormal cholinesterase metabolism and cell apoptosis, among other related regulatory pathways. These results provide a solid theoretical foundation for future in vitro experiments using actual cells and animal experiments to further investigate the application of 6-Gingerol.

## 1. Introduction

Alzheimer’s disease (AD) is the most common form of dementia, characterized by degenerative alterations in the brain’s nervous system. It is typically associated with a decline in cognitive abilities such as learning, memory, and perceptual motor ability. Symptoms include sluggish response, a lack of concentration, progressive memory impairment, etc. [1]. AD has had a significant impact on both the economy and the mental well-being of numerous patients and their families, greatly reducing their quality of life. The presence of intracellular neurofibrillary tangles and the progressive deposition of extracellular Amyloid plaques are typical pathological features of AD [2].

β-amyloid peptide (Aβ) is the main component of Amyloid plaques and plays an important role in the pathophysiological process of AD [3,4]. Based on previous findings, scientists have proposed several popular hypotheses that may induce the development of AD: the hypothesis of Aβ cascade, microtubule-associated protein tau protein (Tau) hyperphosphorylation, cholinergic hypothesis, neuroinflammation, oxidative stress, and metal ions [5]. Oxidative stress and inflammatory response are closely related to Aβ-induced neurotoxicity [6,7]. For example, experiments in vitro/vivo have shown that Aβ has the potential to elevate the levels of reactive oxygen species (ROS) [8,9]. The accumulation of Aβ can lead to oxidative stress reactions [10] and activate inflammatory pathways [11] in the primary neurons in the body. Moreover, lipid peroxidation and mitochondrial DNA oxidation have also been observed in the brains of individuals with AD. Meanwhile, some scholars believe that cell apoptosis [12] and lipid metabolism [13,14] disorders may also lead to the development of AD. Therefore, it is particularly important to explore the mechanisms of preventing and controlling the progression of AD.

After the continuous efforts of researchers, on 6 July 2023, the Food and Drug Administration (FDA) in the United States fully authorized the usage of Leqembi (Lecanemab), an AD treatment drug developed by BIIB. US and ESALY.US [15,16,17]. This also marks the first time in 20 years that the FDA has fully approved a drug for AD. Nevertheless, it is essential to acknowledge that the long-term use of this medication might result in a range of negative responses among patients. While the development of anti-AD drugs has demonstrated certain accomplishments, it is also crucial to consider natural substances present in agricultural products, as they can help alleviate AD symptoms. The similarity between medicine and food offers a more secure and viable alternative. Furthermore, many herbal medicines and functional agricultural products (or food) contain bioactive phytochemical substances with strong antioxidant potential [18,19], such as Allicin in garlic [20], Capsaicin in pepper [21], Curcumin [22], and Gingerol [23] in ginger.

Ginger (Zingiber officinale, Zingiberaceae, Zingiberaceae) is a rhizome plant with both medicinal and culinary uses [19]. It is widely distributed in tropical, sub-tropical, and temperate regions, with an annual global production of 20 million tons [24]. The easy availability of ginger contributes to its high economic value as both an agricultural product and medicine. In traditional Eastern medicine, ginger is recognized for its ability to alleviate head*ACHE*s, nausea, and colds [25]. Modern medicine has also utilized ginger in the treatment of various diseases such as rheumatoid arthritis [26], atherosclerosis, ulcers [27], and depression [28]. Researchers have developed nanomaterials containing ginger’s active ingredients to enhance their effects [29,30]. Among these substances, gingerol is particularly known for its anti-inflammatory and antioxidant properties [31]. Some studies suggest that the active ingredients present in ginger may have the potential to alleviate Aβ-induced neurotoxicity by modulating oxidative stress and inflammatory pathways [32]. However, the research on the specific active compounds in ginger for the treatment of AD is still not sufficiently comprehensive. There have been no reports on identifying the compounds in ginger that are most likely to play a role in AD, and the activated pathways through which these compounds can modulate AD remain unknown.

Network pharmacology, molecular docking, and molecular dynamics are the interdisciplinary field that combines traditional biology, pharmacology, genomics, and other disciplines [33]. They utilize advanced computer analysis technologies, such as multiangle high-throughput screening, to construct a network map of the interactions between drugs, active ingredients, and target diseases [34]. These visualization of data can provide a holistic perspective to explore the holistic relationship between drugs and diseases, and predict the feasibility and mechanism of drug treatment. In recent years, network pharmacology, molecular docking, and molecular dynamics have become a research method and technical means to explore the efficacy of herbal medicine [35,36] or functional agricultural products (food) [37,38], and other related fields. Compared to traditional in vitro and in vivo experiments, this method of evaluating the bioactive function of compounds is efficient, economical, and environmentally friendly, greatly improving the work efficiency of researchers.

This study aimed to investigate the mechanism of the anti-AD effect of active compounds in ginger using network pharmacology, molecular docking, and molecular dynamics simulation techniques. Through network pharmacology and experimental validation in the PC12 cell model constructed using Aβ 1-42, 6-gingerol was identified as the active compound for further research. Molecular simulations were conducted to study the interaction between 6-Gingerol and the top three proteins with the strongest affinity (*ACHE*, *MMP2*, and *PTGS2*), and *CASP3* protein serving as the HUB gene. The results demonstrated a strong binding ability between 6-Gingerol and these disease targets. These findings provide a solid theoretical foundation for future animal experiments and validate the practical application of 6-Gingerol. The complete research process is depicted in Figure 1.

## 2. Materials and Methods

### 2.1. Screening of Active Ingredients in Ginger and Targets for Alzheimer’s Disease

The active ingredients and targets in ginger were screened using the database TCMSP (http://lsp.nwu.edu.cn/tcmsp.php, accessed on 10 May 2023), while AD-related disease targets were screened using multiple databases such as GeneCards (https://www.genecards.org/, accessed on 15 May 2023), OMIM (https://omim.org/, accessed on 15 May 2023), and DisGeNET (https://www.disgenet.org/, accessed on 15 May 2023). To identify key targets, we compared the disease targets and drug targets and identified the intersecting targets.

### 2.2. Functional Enrichment Analysis of Alzheimer’s Disease Targets

The functional enrichment analysis of AD targets was based on Gene Ontology (GO) and Kyoto Encyclopedia of Genes and Genomes (KEGG). Functional annotation in GO entails the attribution of functional terms related to genes from its database, relying on their sequence similarity, experimental evidence, or curated research. The KEGG pathways offer a wide-ranging perspective on the interactions amid genes, proteins, and small molecules across diverse biological processes. The R language package “clusterProfiler” was used to perform GO annotation and KEGG functional enrichment on the key targets. A significance threshold of *p*.adjust < 0.05 was set to identify significantly enriched entries/pathways. The enrichment results were visualized using the R language package “ggplot2”. The genes involved in enriched pathway analysis were extracted, along with their corresponding GO entries and KEGG relationships. A network diagram of the key gene interactions within the pathway was constructed using Cytoscape software.

### 2.3. Agricultural Product–Active Substance–Key Target Network Construction and Identification of HUB Genes

The agricultural product (Ginger), active substances (95), and key targets (95) were used to construct the agricultural product–active substance–key target network using Cytoscape software. To investigate potential interactions among 95 key targets, we utilized STRING (https://string-db.org, accessed on 16 May 2023), a website that constructs a protein–protein interaction (PPI) network for these genes. A medium confidence level of 0.4 was set, and after excluding discrete proteins, the following protein interaction network was obtained. The size of nodes in the network corresponds to their degree of connectivity. Additionally, the top 5 genes with the highest degree of connectivity were identified as the HUB genes.

### 2.4. Cell Experiments

PC12 cells were cultured in RPMI1640 medium (Procell Life Science & Technology Co., Ltd., Wuhan, China) supplemented with fetal bovine serum (FBS, 10%, *v*/*v*, obtained from Thermo Fisher, Waltham, MA, USA), penicillin (100 U/mL, obtained from Thermo Fisher, USA), and streptomycin (100 μg/mL, obtained from Thermo Fisher, USA). The cells were incubated in a humidified atmosphere of 5% CO_2_/95% air at 37 °C. To maintain the cell culture, the PC12 cells were maintained up to 80% confluence in the culture flask and passaged through digestion. Cells used for later experiments were passaged 1~2 times after resuscitation and in logarithmic growth phase, which was smaller than 20 in this study.

Cell viability was evaluated by using Cell Counting Kit-8 (CCK-8, Dojindo, Kyushu, Japan). Before exposure, the PC12 cells were seeded into 96-well plates (8 × 104 cells/mL) for 24 h. Then, the culture medium was removed and replaced with the medium containing imidacloprid and acetamiprid. The cells exposed to different concentrations of Aβ1-42 (Biosynthesis Biotechnology Co., Ltd., Beijing, China) and 6-Gingerol (Acmec Biochemical Co., Ltd., Shanghai, China) were used as the test group, and the control group was treated with 0.1% methanol. The test results were obtained from a microplate reader (Infinite M200 PRO, TECAN, Morrisville, NC, USA).

### 2.5. Molecular Docking Analysis

Molecular docking analysis was conducted using the CB-Dock2 (https://cadd.labshare.cn/cb-dock2/php/index.php, accessed on 20 Dec 2023) platform, to analyze the protein targets of the selected nutrient molecules. This molecular docking tool, CB-Dock2, based on AutoDock Vina analysis, automatically identifies and examines the binding site between the nutrient molecules and the receptor. It simplifies the docking process while enhancing its accuracy. The 2D chemical structure of 6-Gingerol was downloaded from the PubChem (https://pubchem.ncbi.nlm.nih.gov, accessed on 20 Dec 2023) database, and subsequently input into CB-Dock2, where hydrogen atoms (H-atoms) were included, and free energy minimization was performed. Meanwhile, the crystal structures of protein were obtained from RCSB PDB (https://www.rcsb.org/, accessed on 20 Dec 2023), and H-atoms were added, while all water molecules were removed. The binding affinity energy (Vina, kcal/mol) was computed by using CB-Dock2, resulting in the optimal docking model with the lowest energy. The molecular visualization was performed using Pymol (v. 2.5.7) (3D) and Discovery Studio Visualizer (v. 2021) (2D), as the details of ligand–receptor interactions were revealed.

### 2.6. Molecular Dynamics Simulation Analysis

Gromacs2022.3 was utilized for molecular dynamics (MD) simulation. AmberTools22 was employed to incorporate the GAFF force field for preprocessing small molecules, while Gaussian 16W introduced hydrogen atoms and calculated the RESP potential. The obtained potential data were added to the topology file of the MD system. The simulation conditions involved maintaining a static temperature of 300 K and an atmospheric pressure of 1 Bar. The simulation system utilized the Amber99sb-ildn force field and water molecules as the solvent (using the Tip3p water model). To neutralize the total charge of the system, an appropriate number of Na+ ions were added. The simulation system adopted the steepest descent method to minimize the energy. And then, it underwent equilibration under both isothermal isovolumic (NVT) and isothermal isobaric (NPT) ensembles for 100,000 steps, respectively. This coupling constant was set to 0.1 ps, and the duration was 100 ps. Subsequently, a free MD simulation was performed, consisting of 5,000,000 steps with a step length of 2 fs and a total duration of 100 ns. The MD simulations results were visualized using GraphPad Prism Version 9.5 (GraphPad Software, San Diego, CA, USA), including the root mean square deviation (RMSD), root mean square fluctuation (RMSF), the radius of gyration (Rg) value, solvent accessible surface area (SASA), and hydrogen bonds (H-bonds). The Gibbs free energy was calculated based on the RMSD and Rg values via the “bash” script. The Origin (2021, OriginLab, Northampton, MA, USA) software was utilized for generating 3D and 2D Gibbs free energy landscapes.

## 3. Results and Discussion

### 3.1. Active Ingredients in Ginger and the Targets Related to Alzheimer’s Disease

As a unique systematic pharmacology platform for Chinese herbal medicine, TCMSP (http://lsp.nwu.edu.cn/tcmsp.php, accessed on 10 May 2023) allows scholars to discover the relationship between drugs, targets, and diseases. To evaluate the biological information of ginger, “ginger” was used as a keyword for retrieval in the TSCMP database. The filtering criterion was “oral utilization (OB) ≥ 30%”. In order to identify more active substances, this study disregarded the reference condition of “drug similarity (DL) ≥ 0.18”. A total of 143 active ingredients in ginger were found, and are listed in Appendix A. The keyword “Alzheimer’s disease” was used to retrieve information from three databases, GeneCards (https://www.genecards.org/, accessed on 15 May 2023), OMIM (https://omim.org/, accessed on 15 May 2023), and DisGeNET (https://www.disgenet.org/, accessed on 15 May 2023), to obtain the target of AD. Then, the predicted targets from these databases were filtered out. Specifically, the setting parameters “Category” obtained from the Gene Cards database were “Protein Coding” and “Relief”, and Relevance.score was >15. The setting parameter “Score.gda” was >0.3, obtained from the DisGeNET database. A total of 1088 disease targets were obtained by merging the three databases (Figure 2A). Furthermore, the disease targets (1088) and the drug targets (298) were overlapped to obtain 95 intersecting targets defined as the key targets (Figure 2B).

### 3.2. Functional Enrichment Analysis of Key Targets

The R language pack “clusterprofiler” was used to annotate the “GO notes” and KEGG function enrichment of the 95 key targets, and set “*p*.adjust < 0.05” as significantly enriched entries/pathways. A total of 898 BP, 31 CC, 106 MF, and 142 KEGG signaling pathways were obtained. The enrichment results were visualized using the R language pack “ggplot2”. Figure 2C,D show the enriched top 10 “GO notes” results and top 10 KEGG signaling pathways, respectively. It can be seen that these genes are significantly enriched in biological processes, such as responses to xenobiotic stimulus, responses to oxidative stress, responses to toxic substance, lipid and atherosclerosis, diabetic cardiomyopathy, etc. For the exogenous stimulation pathway, there have been relevant reports proving that metal salt ions [39] and acid compounds [40], which can lead to symptoms of AD in mice, are represented by amyloid deposition and delayed movement. These exogenous stimuli can cause the body to experience an imbalance in redox homeostasis, produce inflammatory reactions, and ultimately lead to cell apoptosis. Furthermore, the disruption of lipid homeostasis is associated with AD [41]. In the early stage of AD, there are fatty acid changes in lipid rafts and brain lipid peroxidation levels. Genetic and environmental factors are associated with AD, such as the carrying status of Apolipoprotein E (APOE) and lipid transporters, and dietary lipid content [42].

### 3.3. Construction of Pathway–Key Target Network and Drug–Active Ingredient–Key Target Regulatory Network

The genes in TOP 30 GO (BP:CC:MF = 10:10:10), TOP 10 KEGG, and their corresponding entries and KEGG relationship pairs were extracted. The pathway–key gene interaction network diagrams were constructed using Cytoscape software. As shown in Figure 3, one is composed of 102 points and 387 edges (Figure 3A), and another one is composed of 52 points and 128 edges (Figure 3B). The drug–active ingredient–key target gene network relationship pairs related to 95 key targets were extracted, and the drug–active ingredient–key target interaction network diagram was constructed using Cytoscape software, as shown in Figure 3C. The visual network diagram offers a more intuitive representation of the role of ginger in regulating AD and its associated metabolic pathways. This diagram provides a basis for screening effective active substances in ginger.

### 3.4. Identification of HUB Gene

PPI (protein–protein interaction) networks consist of proteins that engage in diverse life processes, including biological signal transduction, gene expression regulation, energy and substance metabolism, and cell cycle regulation, by interacting with each other. These networks hold immense importance in comprehending the functional connections among proteins [43]. In order to explore the interaction between the 95 key targets, the PPI network of the 95 key target genes was constructed by using the STRING website. The confidence was 0.4 (medium confidence = 0.4). After removing discrete proteins, the following PPI network was obtained (Figure 4). The size of nodes is directly proportional to the degree (the specific information is shown in Appendix A). The color of nodes from white to red and the increase of nodes indicates that the degree is gradually increasing. The Top 5 degree genes are HUB genes (*ALB*, *ACTB*, *GADPH*, *CASP3* and *CAT*). These HUB genes were subsequently used in molecular docking and molecular dynamics studies of compound protein interactions.

### 3.5. Screening of Characteristic Active Ingredients in Ginger and Validation via PC12 Cell Damage Model Constructed Using Aβ1-42

The above network pharmacology results (Appendix A), along with our laboratory’s previous screening and identification of characteristic active substances in ginger, constitute the key evidence in screening compounds. This evidence suggests that 6-Gingerol is an effective active substance for treating AD. Furthermore, several studies in the literature have also highlighted the significance and role of 6-Gingerol in AD research [44,45]. Therefore, 6-Gingerol was used in the study of cell models.

Aβ 1-42 is known for cytotoxic effects and is commonly used as an in vitro cell model for studying AD in neural cells, such as PC12 and Neuro-2a [46]. Numerous research reports suggested that Aβ 1-42 negatively affects neuronal apoptosis, redox levels imbalance, and inflammatory response [47]. In this study, different concentrations of Aβ 1-42 were used. After co-incubation with PC12 cells for 24 h, the cells were further incubated in an environment with a concentration of 10 μM Aβ 1-42 for 24 h (Figure 5A). In terms of cell viability, it was observed that 6-Gingerol had a certain improvement effect on PC12 cells incubated with Aβ 1-42 for 24 h (Figure 5B). Based on these results, subsequent research involved conducting molecular simulations and molecular dynamics experiments using 6-Gingerol as the active substance.

### 3.6. Molecular Docking Analysis

A molecular docking strategy is employed in computer-aided drug research to investigate the interaction between nutrient molecules and protein, as well as predict binding modes and affinities. The CB-Dock2 platform was used for molecular docking analysis to evaluate the binding affinity between 6-Gingerol and various proteins including *ACHE*, *ADRB2*, *BAX*, *BCL2*, *CASP3*, *CASP9*, *ESR1*, *JUN*, *MMP2*, *MMP9*, *PPARG*, *PTGS2*, *RELA*, *SLC6A2*, *SLC6A3*, and *SLC6A4*. A lower docking energy indicates a stronger binding affinity between the protein and nutrient molecules, according to the docking hypothesis. The Vina score (shown in Table 1) was calculated as an indicator of binding affinity. The top three lowest docking energies were observed for the combinations of *ACHE* (4m0e)-6-Gingerol, *MMP2* (1gxd)-6-Gingerol, and *PTGS2* (5f19)-6-Gingerol. Furthermore, it is noteworthy that *CASP3* emerged as a HUB gene for AD in the results presented in Figure 4.

According to Figure 6A, 6-Gingerol interacted with *ACHE* (4m0e) via van der Waals to ALA204, GLY121, ASP74, ASN87, LEU130, GLY126, SER125, GLY120, HIS447, TRP286, PHE297, and PHE295. Meanwhile, 6-Gingerol displayed stable binding to *ACHE* via hydrogen bonding with SER203, GLY122, TYR337, TYR124, and TRP86. Finally, the stable binding of 6-Gingerol to *ACHE* involved pi-pi stacked and pi-alkyl interaction bonds with residues TRP86, PHE338, and TYR341. As shown in Figure 6B, 6-Gingerol interacted with *CASP3* (4ps0) via van der Waals to THR77, ASN80, LEU81, TYR276, PHE275, GLN225, ALA227, and TYR226. Meanwhile, 6-Gingerol displayed stable binding to *CASP3* via hydrogen bonding with HIS277, LYS224, LEU223, and ASP228. Finally, the stable binding of 6-Gingerol to *CASP3* involved alkyl and pi-alkyl interaction bonds with residues LYS82 and MET44. As shown in Figure 6C, 6-Gingerol interacted with *MMP2* (1gxd) via van der Waals to GLY74, ALA375, HIS164, PHE402, THR399, ALA390, PRO388, THR397, TYR396, and ARG72. Meanwhile, 6-Gingerol displayed stable binding to *MMP2* via hydrogen bonding with ARG462, GLY389, ILE395, ALA393, and PRO394. Finally, the stable binding of 6-Gingerol to *MMP2* involved alkyl and pi-alkyl interaction bonds with residues ALA163, CYS73, VAL371, LEU162, LEU391, HIS374, LEU370, and LEU479. As shown in Figure 6D, 6-Gingerol interacted with PTGS2 (5f19) via van der Waals to LEU534, PHE529, TYR348, ALA527, VAL349, SER353, MET522, TRP387, LEU384, TYR385, VAL344, PHE381, GLY526, ASN375, ILE377, PHE205, GLY227, and GLY533. Meanwhile, 6-Gingerol displayed stable binding to PTGS2 via hydrogen bonding with LEU531. Finally, the stable binding of 6-Gingerol to PTGS2 involved pi-pi stacked, alkyl, pi-sigma, and pi-alkyl interaction bonds with residues PHE518, VAL523, VAL228, PHE209, and LEU352. The findings demonstrate a favorable binding activity between 6-Gingerol and the targets.

### 3.7. Molecular Dynamics Simulation Analysis

Furthermore, a 100 ns MD simulation was conducted to investigate the dynamic properties of the *ACHE*–6-Gingerol, *CASP3*–6-Gingerol, *MMP2*–6-Gingerol, and PTGS2–6-Gingerol complexes obtained from molecular docking. The results of the MD simulations, particularly the root mean square deviation (RMSD), root mean square fluctuation (RMSF), the radius of gyration (Rg) value, solvent accessible surface area (SASA), and hydrogen bonds (H-bonds), provide a crucial role in determining the stability of chemical compounds and protein complexes. Additionally, these simulations also help in evaluating the stability of protein tertiary structures after the binding of small molecules, the hydrophobicity of amino acid residues, and other relevant factors.

The RMSD was utilized to examine the mobility of the receptor–ligand complex, showing protein conformational changes [48]. The RMSF can be utilized to illustrate variations in the complex at the residue level [49]. The Rg value shows the degree of system constraint and binding tightness, as well as reflecting protein folding [50]. The Rg value is a measurement linked to the tertiary structure of protein volume that has been used to gain insight into a protein’s stability in a biological system. A higher Rg value suggests a better probability of producing flexible ligands. As a result, the bigger the Rg number, the less stable the system. In contrast, a lower Rg number indicates a dense and tightly packed structure. The RMSD and Rg values were used to build a Gibbs free energy 3D and 2D landscape in order to find and analyze its steady-state properties. The Gibbs free energy landscape depicts the stability of the receptor–ligand combination [51]. The blue and purple hues in the landscape indicate the region where the complex’s stable conformation is found at a lower energy level.

#### 3.7.1. Molecular Dynamics Simulation of the *ACHE*–6-Gingerol Complex

As shown in Figure 7A, the RMSD curve of 6-Gingerol (red line) was stabilized around 0.18 nm for 0–70 ns. The RMSD curve of *ACHE* (blue line) was relatively stable at 0.16–0.18 nm for 60–100 ns. The RMSD curves of the *ACHE*–6-Gingerol complex (green line) were generally steady around 0.24–0.26 nm for 60–100 ns, indicating that 6-Gingerol could generate a stable complex system with *ACHE*. As presented in Figure 7B, the residues 70–100, 255–270, 380–390, and 490–500 of *ACHE* had greater residue flexibility. Figure 7C demonstrates that the Rg value of *ACHE* remained steady for 30–100 ns, with the lowest mean Rg value ranging between 2.32 and 3.34 nm. Hydrogen bonding is a strong non-covalent interaction. The complex systems of *ACHE*–6-Gingerol with stable hydrogen bonds (1–3) were shown to exist stably throughout the MD simulation. Meanwhile, the maximum number of hydrogen bonds in the *ACHE*–6-Gingerol complex was 5 (Figure 7D). The hydrogen bond that exists between the ligand and the receptor contributes to the complex’s stabilization. The 3D and 2D Gibbs energy landscape of the *ACHE*–6-Gingerol is shown in Figure 7E,F. The *ACHE*–6-Gingerol complex was in a low Gibbs free energy when the Rg value was 2.305–2.33 nm and the RMSD value was 0.15–0.20 nm.

Acetylcholinesterase (*ACHE*) is the most widely reported target related to AD in previous studies. Cholinergic neurotransmission disorders are well established in AD [52]. Some studies have shown reduced *ACHE* in AD patients, while elevated levels of *ACHE* variants T30 and T14 have been found [53]. It is worth noting that lipase and cholinesterase belong to the same class of enzymes [54]. There are also numerous reports linking changes in lipid metabolism, live cells, apoptotic cells, and the distribution of necrotic brain cells to the activity of acetylcholinesterase in serum and brain [55]. Restigmine, a drug targeting *ACHE*, is considered as one of the treatment options for AD [56]. Impaired lipid metabolism, which is associated with decreased acetylcholine levels and alterations in phospholipid metabolism, is a key characteristic of AD [57]. Combining MD simulations, it can be inferred that 6-Gingerol has a regulatory effect on AD.

#### 3.7.2. Molecular Dynamics Simulation of the *CASP3*-6-Gingerol Complex

The RMSD curve of 6-Gingerol (blue line) stabilized around 0.4 nm for 12–28 ns and 0.3 nm for 4–50 ns (Figure 8A**)**. Similarly, the RMSD curve of *CASP3* (purple line) remained relatively stable at 0.3 nm for 12–100 ns. The RMSD curves of the *CASP3*-6-Gingerol complex (green line) indicated a steady around 0.3–0.4 nm for 25–62 ns, suggesting the formation of a stable complex system between *CASP3* and 6-Gingerol. The residues 174–189 of *CASP3* exhibited greater flexibility, as depicted in Figure 8B. The Rg value of *CASP3* remained constant at 1.85–1.88 nm for 16–100 ns, as illustrated in Figure 8C. The *CASP3*-6-Gingerol complex demonstrated stable hydrogen bonds for 0–80 ns, with a maximum of four hydrogen bonds. These hydrogen bonds contribute to the stability of the complex. Figure 8E,F present the 3D and 2D Gibbs energy landscape of the *CASP3*-6-Gingerol complex, respectively. The complex exhibited low Gibbs free energy when the Rg value was 1.80–1.83 nm and the RMSD value was 0.30–0.355 nm.

Caspase-3 (*CASP3*), a cytoplasmic protease, plays a crucial role in apoptosis [58]. Research has shown that *CASP3* is involved in the cleavage of β-amyloid protein, as well as Tau protein and presenilin protein (PS1, PS2), which are closely associated with AD [59]. Crocin significantly reduces *CASP3* protein expression in cell models of Alzheimer’s disease, offering promising potential for treating the disease by decreasing the cell apoptosis rate and alleviating mitochondrial dysfunction [60]. The strong binding and favorable kinetic index of 6-Gingerol to *CASP3* highlight the potential effectiveness of this natural product in regulating AD.

#### 3.7.3. Molecular Dynamics Simulation of the *MMP2*-6-Gingerol Complex

As depicted in Figure 9A, the RMSD curve of 6-Gingerol (blue line) remained stable at around 0.2 nm throughout the simulations. The RMSD curve of *MMP2* (red line) was relatively stable at 0.4–0.42 nm for 20–100 ns. The RMSD curves of the *MMP2*-6-Gingerol complex (green line) generally showed stability around 0.42–0.46 nm for 20–100 ns, suggesting the formation of a stable complex system between 6-Gingerol and *MMP2*. Figure 9B highlights that the residues 80–90, 270–300, and 420–480 of *MMP2* displayed greater flexibility. Figure 9C demonstrates a steady Rg value of 2.80–2.85 for *MMP2* during 15–60 ns. The *MMP2*-6-Gingerol complex exhibited stable hydrogen bonds (1–4) throughout the MD simulation, with a maximum of six hydrogen bonds (Figure 9D). These hydrogen bonds contribute to the stabilization of the ligand–receptor complex. The 3D and 2D Gibbs energy landscape of the *MMP2*-6-Gingerol complex are illustrated in Figure 9E,F. The complex showed low Gibbs free energy when the Rg value was 2.77–2.83 nm and the RMSD value was 0.35–0.45 nm.

Matrix Metalloproteinase-2 (*MMP2*) is a zinc-dependent protease that primarily targets extracellular matrix proteins. It is a neutral enzyme that plays a crucial role in regulating proteolytic activity. The dysregulation of MMP2 has been linked to AD [61]. This suggests that the accumulation of MMP2 in neurofibrillary tangles may be a protective response aimed at reducing the production of toxic truncated tau species in the brains of individuals with AD [62]. After MD simulation, it was found that 6-Gingerol has a very nice binding ability with *MMP2*, indicating that 6-Gingerol has good prospects in the treatment of AD.

#### 3.7.4. Molecular Dynamics Simulation of the *PTGS2*-6-Gingerol Complex

As shown in Figure 10A, the RMSD curve of 6-Gingerol (orange line) remained stable at approximately 0.8 nm throughout the simulations. The RMSD curve of *PTGS2* (red line) was relatively stable at 0.18–0.20 nm for 40–100 ns. The RMSD curves of the *PTGS2*-6-Gingerol complex (purple line) were generally steady around 0.22–0.24 nm for 40–100 ns, indicating that 6-Gingerol might generate a stable complex system with *PTGS2*. As presented in Figure 10B, the residues 54–56 and 130–150 of *PTGS2* had greater residue flexibility. Figure 10C demonstrates that the Rg value of *PTGS2* remained steady at 2.46–2.48 nm for 25–100 ns. The MD simulation revealed the presence of stable hydrogen bonds (1–2) in the *PTGS2*-6-Gingerol complex. Meanwhile, the maximum number of hydrogen bonds in the *PTGS2*-6-Gingerol complex was three (Figure 10D). The hydrogen bond that exists between the ligand and the receptor contributes to the complex’s stabilization. Figure 10E,F depict the 3D and 2D Gibbs energy landscape of the *PTGS2*-6-Gingerol complex. The complex exhibited low Gibbs free energy when the Rg value ranged from 2.81 to 2.85 nm and the RMSD value was between 0.4 and 0.5 nm.

Prostaglandin-endoperoxide synthase 2 (*PTGS2*), also referred to as cyclooxygenase 2 (*COX-2*), is an enzyme that acts as the rate-limiting factor in the synthesis of prostaglandins (*PGs*). It performs crucial functions during the initial stages of pregnancy [63]. Elevated levels of *PTGS2/COX-2* and *PGs* play a role in the development of AD. AD is characterized by the buildup of Aβ protein and tau hyperphosphorylation, and the relevant mechanisms and pathways may include neuroinflammation, oxidative stress, synaptic plasticity, neurotoxicity, autophagy, and apoptosis [64]. Due to the close binding ability of 6-Gingerol and *PTGS2* in the molecular docking and MD experiments, 6-Gingerol can be considered as a potential compound for the treatment of AD.

*ACHE*, *MMP2*, *PTGS2*, and *CASP3* are all targets associated with AD, and *ACHE* is the most widely reported target in related research. The molecular simulation and MD results confirmed that 6-Gingerol may be regulated by *ACHE*, *MMP2*, *PTGS2*, and *CASP3* in improving the progression of AD.

## 4. Conclusions

This study combines network pharmacology, molecular docking, and molecular dynamics simulation technology to screen out the most likely active substance, 6-Gingerol, in regulating AD from the various compounds in ginger. It aims to predict, verify, and understand the related mechanisms of 6-Gingerol. Initially, the TCMSP database was used to predict 298 targets related to 143 active ingredients in ginger. Subsequently, by merging data from the GeneCards, OMIM, and DisGeNET databases, 1088 AD-related targets were obtained. The overlap between disease targets (1088) and drug targets (298) resulted in 95 cross-targets, which were defined as critical targets. A functional enrichment analysis, including GO annotation and KEGG pathway analysis, was performed on these 95 targets. The pathways in which the active substances in ginger may be involved in regulating AD include response to exogenous stimuli, response to oxidative stress, toxic substance reactions, lipids and atherosclerosis, diabetic cardiomyopathy, etc. Furthermore, the drug–active ingredient–key target interaction network diagram was constructed using Cytoscape. This diagram showcases the relationship between active ingredients and key targets, providing a more intuitive representation of ginger’s role in regulating AD and its associated metabolic pathways. The findings of this study serve as a basis for screening effective active substances in ginger. Additionally, PPI networks were developed to identify HUB genes (*ALB*, *ACTB*, *GAPDH*, *CASP3*, and *CAT*) through careful screening. Based on the results of network pharmacology and laboratory screening, 6-Gingerol was selected as the active ingredient for studying its effects on PC12 cells. The molecular dynamics studies focused on the top three proteins (*ACHE*, *MMP2*, and *PTGS2*) with the strongest binding affinity, along with the *CASP3* protein as the HUB gene. The results indicate that 6-Gingerol exhibits strong binding ability to these disease targets, suggesting its potential role in regulating Alzheimer’s disease at the molecular level and cholinesterase metabolism abnormalities, cell apoptosis, and other related regulatory pathways. These findings provide a solid theoretical foundation for future in vitro and animal experiments involving the application of 6-Gingerol.

## Figures and Tables

**Figure 1 foods-13-00612-f001:**
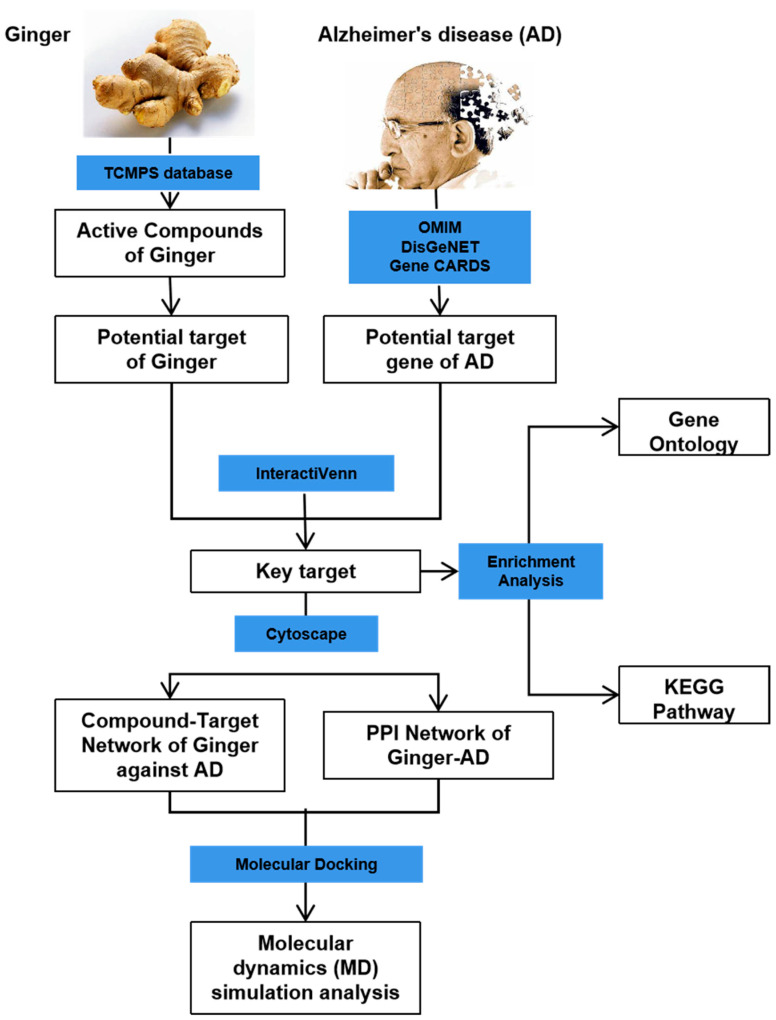
Flow chart showing the experimental design.

**Figure 2 foods-13-00612-f002:**
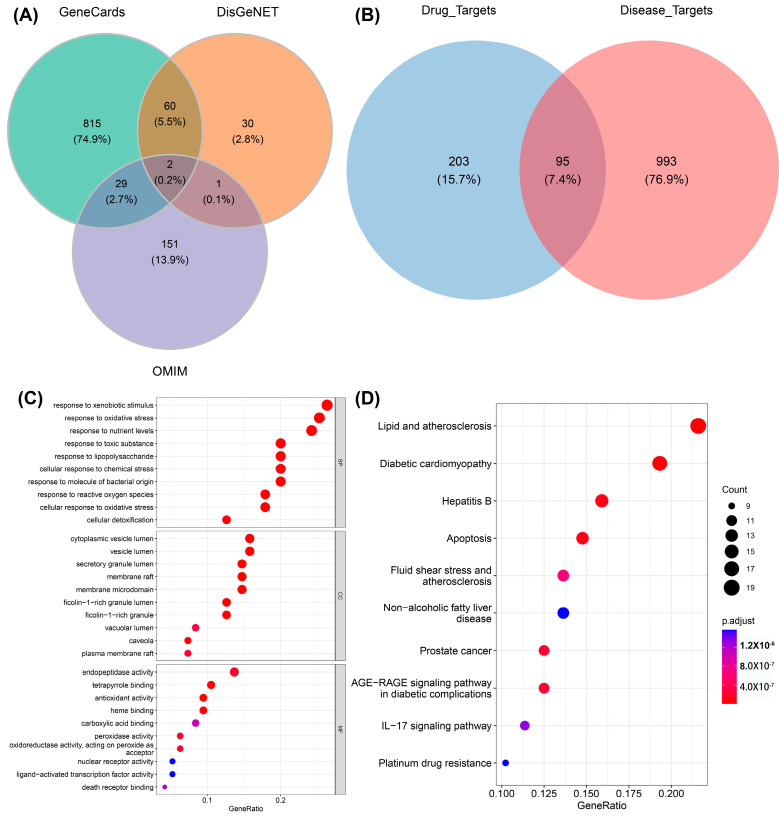
The involvement of active compounds in related pathways in AD. (**A**) Venn diagram showing the relevant targets in AD. (**B**) Venn diagram showing the disease targets (1088) alongside the targets (298) associated with ginger active constituents. (**C**) GO enrichment results for disease targets (top 10 for BP, CC, and MF, respectively; 30 entries in total). (**D**) KEGG enrichment notes for disease targets (top 10). The size of different circles in the bubble chart represents different gene numbers, and different colors represent different *p*.adjust. The color from blue to red indicates a strong degree of significance in (**C**,**D**).

**Figure 3 foods-13-00612-f003:**
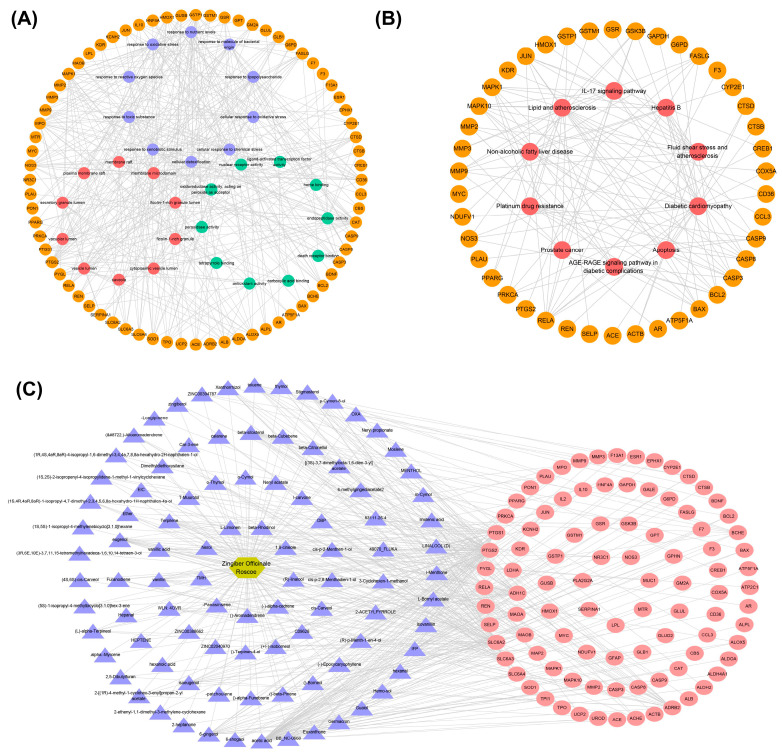
Construction of networks. (**A**) Pathway–key gene interaction network for GO. The orange circle represents the key gene, and the purple, red, and green circles represent Top 10 in BP, Top 10 in CC, and Top 10 in MF, respectively. (**B**) Pathway–key gene interaction network for KEGG. The orange circle represents the key genes, and the red represents the Top 10 in KEGG signaling pathways. (**C**) The regulatory network of drug–active ingredient–key target regulatory network (191 nodes, 444 edges). The golden hexagon represents one kind of traditional Chinese medicine, the purple triangles represent the active ingredients, and the red ellipses represent the key targets.

**Figure 4 foods-13-00612-f004:**
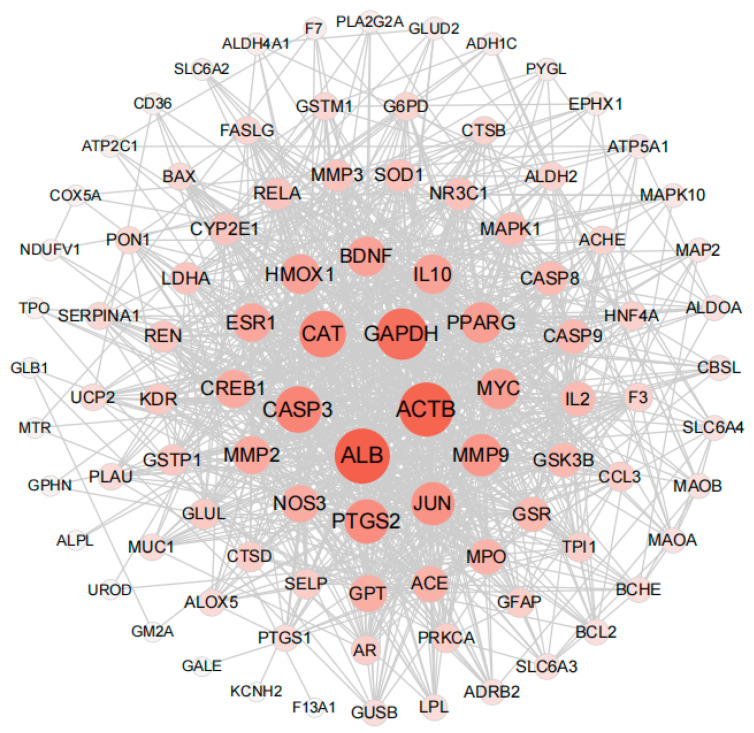
Protein interaction network analysis results of key targets (95 nodes, 958 edges).

**Figure 5 foods-13-00612-f005:**
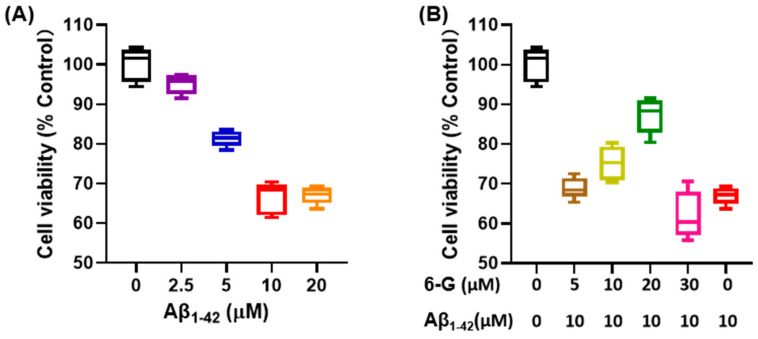
Cell viability test using the CCK-8 kit. (**A**) PC12 cells exposed to different concentrations of Aβ1-42 culture medium. (**B**) Different concentrations of 6-Gingerol are used to regulate a PC12 cell injury model induced by Aβ1-42.

**Figure 6 foods-13-00612-f006:**
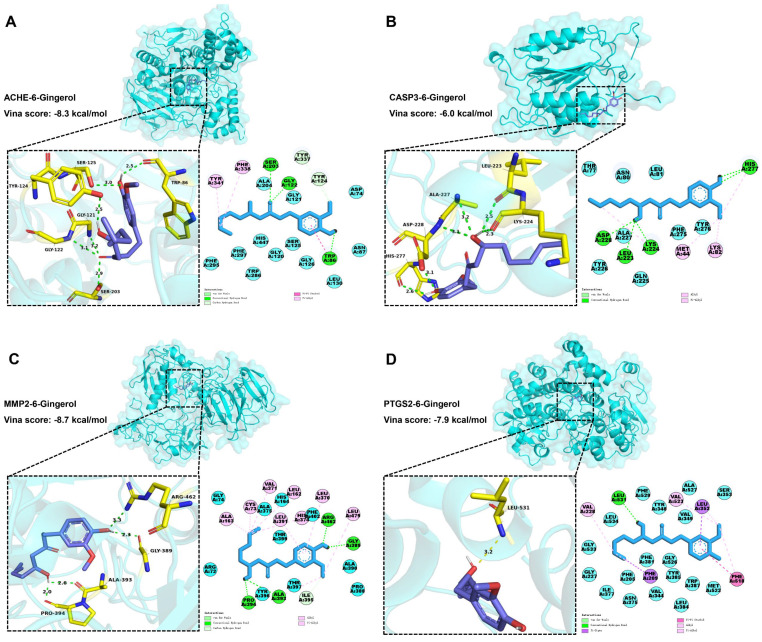
Molecular docking results of 6-Gingerol and the targets. (**A**) Docking model of *ACHE*–6-Gingerol with the lowest binding affinity (−8.3 kcal/mol). (**B**) Docking model of *CASP3*–6-Gingerol with the lowest binding affinity (−6.0 kcal/mol). (**C**) Docking model of *MMP2*–6-Gingerol with the lowest binding affinity (−8.7 kcal/mol). (**D**) Docking model of PTGS2–6-Gingerol with the lowest binding affinity (−7.9 kcal/mol).

**Figure 7 foods-13-00612-f007:**
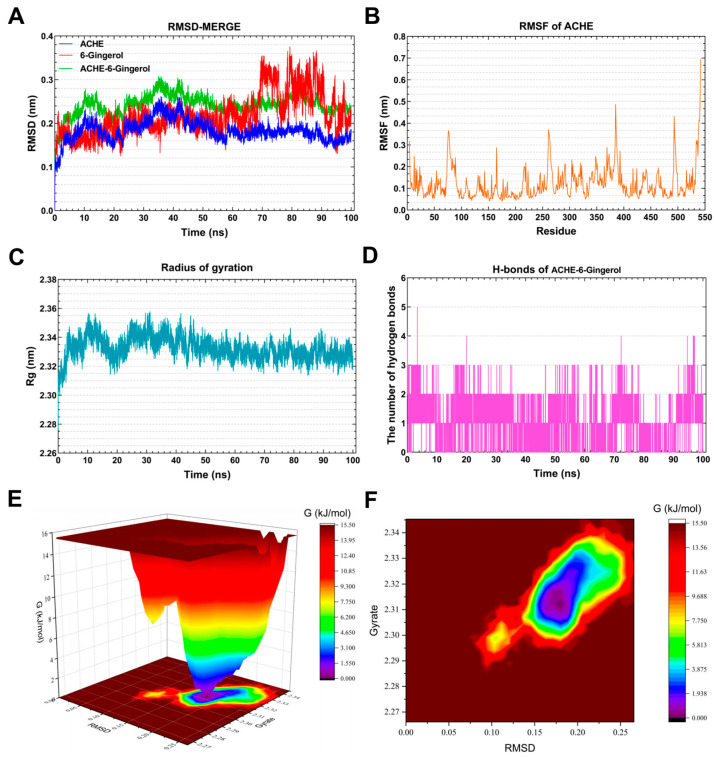
Molecular dynamics simulation analysis of the *ACHE*-6-Gingerol complex. (**A**) RMSD curve of 6-Gingerol (red line), *ACHE* (blue line), and the *ACHE*–6-Gingerol complex (green line). (**B**) RMSF curve of *ACHE*. (**C**) Rg curve of *ACHE*. (**D**) Hydrogen bonds of the *ACHE*–6-Gingerol complex. (**E**,**F**) Three-dimensional and two-dimensional Gibbs free energy landscape of the *ACHE*–6-Gingerol complex.

**Figure 8 foods-13-00612-f008:**
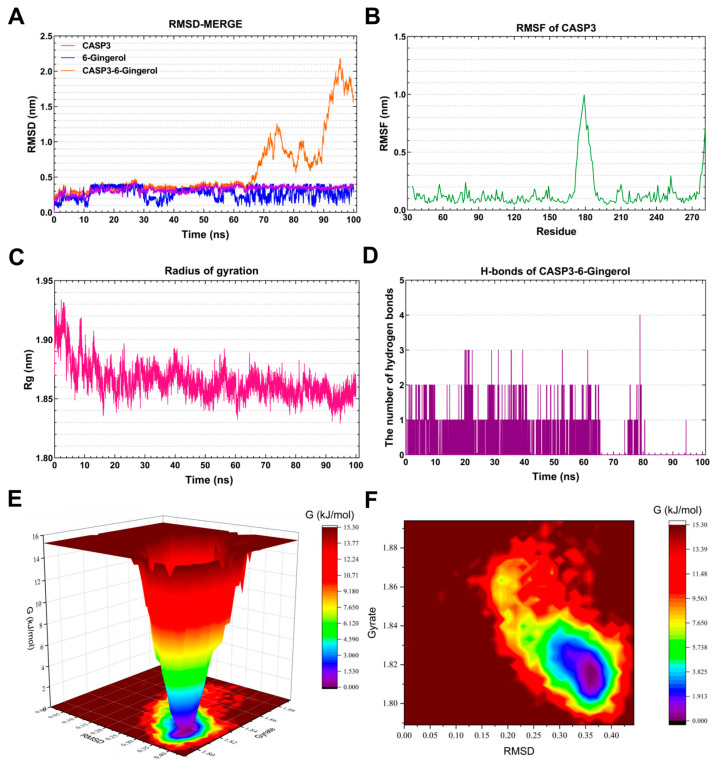
Molecular dynamics simulation analysis of the *CASP3*-6-Gingerol complex. (**A**) RMSD curve of 6-Gingerol (blue line), *CASP3* (purple line), and the *CASP3*–6-Gingerol complex (orange line). (**B**) RMSF curve of *CASP3*. (**C**) Rg curve of *CASP3*. (**D**) Hydrogen bonds of the *CASP3*–6-Gingerol complex. (**E**,**F**) Three-dimensional and two-dimensional Gibbs free energy landscape of the *CASP3*–6-Gingerol complex.

**Figure 9 foods-13-00612-f009:**
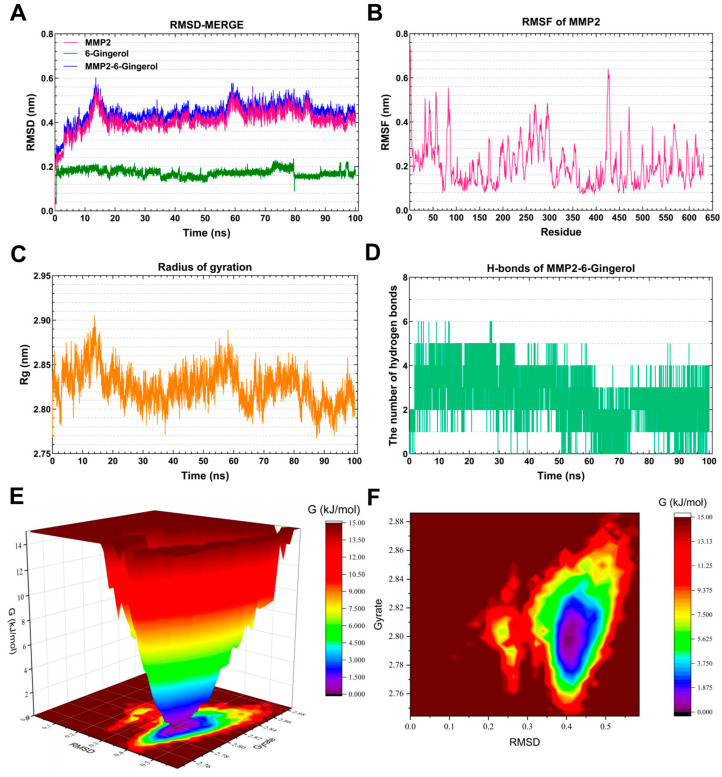
Molecular dynamics simulation analysis of the *MMP2*-6-Gingerol complex. (**A**) RMSD curve of 6-Gingerol (green line), *MMP2* (red line), and the *MMP2*–6-Gingerol complex (blue line). (**B**) RMSF curve of *MMP2*. (**C**) Rg curve of *MMP2*. (**D**) Hydrogen bonds of the *MMP2*–6-Gingerol complex. (**E**,**F**) Three-dimensional and two-dimensional Gibbs free energy landscape of the *MMP2*–6-Gingerol complex.

**Figure 10 foods-13-00612-f010:**
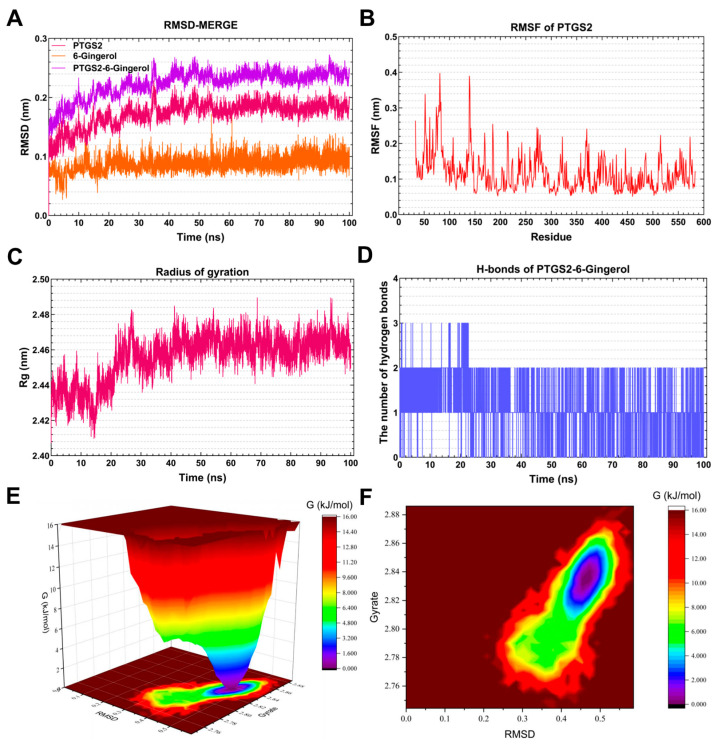
Molecular dynamics simulation analysis of the *PTGS2*-6-Gingerol complex. (**A**) RMSD curve of 6-Gingerol (orange line), *PTGS2* (red line), and the PTGS2–6-Gingerol complex (purple line). (**B**) RMSF curve of *PTGS2*. (**C**) Rg curve of *PTGS2*. (**D**) Hydrogen bonds of the *PTGS2*–6-Gingerol complex. (**E**,**F**) Three-dimensional and two-dimensional Gibbs free energy landscape of the *PTGS2*–6-Gingerol complex.

**Table 1 foods-13-00612-t001:** Vina scores of simulated binding between 6-Gingerol and various proteins.

Target	PDB ID/Alphafold ID	Vina Score (kcal/mol)
*MMP2*	1gxd	−8.7
*ACHE*	4m0e	−8.3
*PTGS2*	5f19	−7.9
*SLC6A3*	AF-Q01959-F1	−7.7
*ADRB2*	3sn6	−7.3
*MMP9*	1l6j	−7.3
*SLC6A4*	7lia	−7.2
*SLC6A2*	AF-P23975-F1	−7.1
*ESR1*	5kra	−6.8
*CASP9*	5wvc	−6.3
*PPARG*	3e00	−6.2
*CASP3*	4ps0	−6.0
*BCL2*	5jsn	−5.8
*BAX*	4bdu	−5.4
*RELA*	1nfi	−5.2
*JUN*	AF-P05412-F1	−4.6

## Data Availability

The original contributions presented in the study are included in the article and supplementary materials, further inquiries can be directed to the corresponding author.

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
