# Peer review of "Screening of Active Substances Regulating Alzheimer’s Disease in Ginger and Visualization of the Effectiveness on 6-Gingerol Pathway Targets"

_foods, 2024, doi:10.3390/foods13040612_

Round 1
Reviewer 1 Report
Comments and Suggestions for Authors
Mainly theoretical manuscript based on simulations and database searches. The authors presented obtained data within numerous figures and tables, and determined main components of ginger that may be corelated with drug targets in curing AD. However, beside simulations and search, only few experiments have been performed that show slight influence of the main detected component of ginger that may influence AD. Experiments were performed on PC12 model cells. Although there are no significant remarks regarding the manuscript, in my opinion it still lack some more detailed explanations that will further justify why those simulations and database searches are important for detecting drugs for specific diseases. This may be given as additional paragraph in Introduction section, as well as additional comments within appropriate sections as indicated in PDF document.

Minor English language corrections needed, mainly confused verbs and nouns. Check out with some software as Grammarly or similar.
Author Response
Dear reviewer,
Thank you for your comments concerning our manuscript entitled “Screening of active substances regulating Alzheimer's disease in ginger and visualization of the effectiveness on 6-Gingerol pathway targets” (Submission ID: foods-2855269). The comments and suggestions were highly insightful and enabled us to greatly improve the quality of our manuscript. We have studied comments carefully and have made correction which we hope meet with approval. Revisions in the text are shown using red color for additions or changes. The main corrections in the paper and the point-by-point response to reviewers’ comments are as following. Thank you for your suggestions. We have modified the manuscript based on your valuable modifications point by point in revised manuscript.
Major Comments
Mainly theoretical manuscript based on simulations and database searches. The authors presented obtained data within numerous figures and tables, and determined main components of ginger that may be corelated with drug targets in curing AD. However, beside simulations and search, only few experiments have been performed that show slight influence of the main detected component of ginger that may influence AD. Experiments were performed on PC12 model cells. Although there are no significant remarks regarding the manuscript, in my opinion it still lack some more detailed explanations that will further justify why those simulations and database searches are important for detecting drugs for specific diseases. This may be given as additional paragraph in Introduction section, as well as additional comments within appropriate sections as indicated in PDF document.
Response: We appreciate your insightful and constructive comments, which can help us make the manuscript more comprehensive. It is indeed necessary for us to explain in the manuscript why using database search and molecular simulation methods is important for detecting drugs for specific diseases. We have carefully modified and responded to the places you marked in the PDF manuscript point by point.
Detailed Comments
- In Abstract, who developed...identifed, why developed...identified? it is unclear what authors want to say by this sentence, please rewrite.who select...conduct, why select conduct? it is unclear what authors want to say with this sentence, please rewrite.
Response: Thank you for your suggestion. We have made modifications to the abstract section as follow:
Abstract: Ginger has been reported to potentially treat Alzheimer's disease (AD), but the specific compounds responsible for this biological function and their mechanisms are still unknown. In this study, a combination of network pharmacology, molecular docking, and dynamic simulation technology was used to screen active substances that regulate AD and explore their mechanisms. The TCMSP, GeneCards, OMIM, and DisGeNET databases were utilized to obtain 95 cross-targets related to ginger's active ingredients and AD as key targets. Functional enrichment analysis revealed that the pathways in which ginger's active substances may be involved in regulating AD include response to exogenous stimuli, response to oxidative stress, response to toxic substances, and lipid metabolism, among others. Furthermore, a drug-active ingredient-key target interaction network diagram was constructed, highlighting that 6-Gingerol is associated with 16 key targets. Additionally, a protein-protein interaction (PPI) network was mapped for the key targets, and HUB genes (ALB, ACTB, GAPDH, CASP3, and CAT) were identified. Based on the results of network pharmacology and cell experiments, 6-Gingerol was selected as the active ingredient for further investigation. Molecular docking was performed between 6-Gingerol and its 16 key targets, and the top three proteins with the strongest binding affinities (ACHE, MMP2, and PTGS2) were chosen for molecular dynamics analysis together with the CASP3 protein as the HUB gene. The findings indicate that 6-Gingerol exhibits strong binding ability to these disease targets, suggesting its potential role in regulating AD at the molecular level, as well as in abnormal cholinesterase metabolism and cell apoptosis, among other related regulatory pathways. These results provide a solid theoretical foundation for future in vitro experiments using actual cells and animal experiments to further investigate the application of 6-Gingerol.
- On line 101-102. Why this proteins were used for analysis? Please explain.
Response: The question you asked is very meaningful. These proteins are the three most tightly bound proteins obtained after molecular docking of 6-gingerol and the 16 key proteins in Table 1, and the proteins related to 6-gingerol involved in the HUB gene.
- On line 103-104. Please add that PC12 cells were used for analysis and explain why those cell were depicted.
Response: Thank you for you suggestion. We have added a description of cell experiments in this section as follow.
This study aimed to investigate the mechanism of the anti-AD effect of active compounds in ginger using network pharmacology, molecular docking, and molecular dynamics simulation techniques. Through network pharmacology and experimental validation in the PC12 cell model constructed by Aβ 1-42, 6-gingerol was identified as the active compound for further research. Molecular simulations were conducted to study the interaction between 6-Gingerol and the top three proteins with the strongest affinity (ACHE, MMP2, and PTGS2), and CASP3 protein serving as the HUB gene. The results demonstrated a strong binding ability between 6-Gingerol and these disease targets. These findings provide a solid theoretical foundation for future animal experiments and validate the practical application of 6-Gingerol. The complete research process is depicted in Figure 1.
PC12 cells are the valuable tool, because they exposed to Aβ1-42 are more commonly used to create cellular models of Alzheimer's disease. In addition, the test period of in vitro cells is short and the experimental cost is low. The effect of ginger active substances on the treatment of Alzheimer's disease can be quickly known.
[References]
Zheng Y; Xu G; Ni Q; et al. Microemulsion Delivery System Improves Cellular Uptake of Genipin and Its Protective Effect against Aβ1-42-Induced PC12 Cell Cytotoxicity. Pharmaceutics 2022, 14(3), 617.
- On line 140-149. unfinished sentence, please finish.
Response: Thank you for your suggestion. We have merged and revised sections 2.4 and 2.5 in the new manuscript as follow.
2.4. Cell experiments
PC12 cells were cultured in RPMI1640 medium (Procell Life Science & Technology Co., Ltd. ,Wuhan, China) supplemented with fetal bovine serum (FBS, 10%, v/v, obtained from Thermo Fisher, USA), penicillin (100 U/mL, obtained from Thermo Fisher, USA) and streptomycin (100 μg/mL, obtained from Thermo Fisher, USA). The cells were incubated in a humidified atmosphere of 5% CO2/95% air at 37 °C. To maintain the cell culture, PC12 cells were maintained up to 80% confluence in the culture flask and passaged through digestion. Cells used for later experiment were passaged 1~2 times after resuscitation and in logarithmic growth phase, and smaller than 20 in this study.
Cell viability was evaluated by using Cell Counting Kit-8 (CCK-8, Dojindo, Japan). Before exposured, PC12 cells were seeded into 96-well plates (8×104 cells/mL) for 24 hours. Then, the culture medium was removed and replaced by the medium containing imidacloprid and acetamiprid. The cells exposed to different concentrations of Aβ1-42 (Biosynthesis Biotechnology CO.Ltd, Beijing, China) and 6-Gingerol (Acmec Biochemical Co., Ltd, Shanghai, China)was as the test group, and the control group was treated with 0.1% methanol. The test results were obtained from microplate reader (Infinite M200 PRO, TECAN, USA).
Your issue has been resolved resolved.
- In the section 3.7:After this "flood of data" author need to give some comments on what this data suggest about gingerol and AD, and why this data are important to this relation, since this is the topic of the manuscript./ Same as previous comment./ Same as previous two. There are many data here given and authors need to connect presented data to explain why are they given here and what is relation between molecular docking / dynamics and curring AD with gingerol./ Same.
Response: Thanks for this comment. We have thoroughly checked this section for language grammar errors and ensured the consistency of the results. In addition, add some latest reference materials to support the results. There are some modifications involved, and they have been made in the manuscript.
Comments on the Quality of English Language
Minor English language corrections needed, mainly confused verbs and nouns. Check out with some software as Grammarly or similar.
Response: Thank you very much. We have revised the language and grammar sections of the article according to your suggestions. There are some modifications involved, and they have been made in the manuscript.
Thanks a lot for your comments again. We truly appreciate it that enhances our work to a great extent. Without your help, our conclusion is weak.
Sincerely,
Author

Reviewer 2 Report
Comments and Suggestions for Authors
1. The paper titled “Screening of active substances regulating Alzheimer's disease in ginger and visualization of the effectiveness on 6-Gingerol 3 pathway targets” described the role of practical application of 6-Gingero using molecular studies for the Alzheimer's disease using a series of procedure for target compounds related to active ingredients and prediction using TCMSP database along with disease obtained through GeneCards, OMIM, and DisGeNET databases. The manuscript has potential but need following changes before consideration
- Title is good and elaborative
- Abstract is written poor as the methodology “In this study, 298 targets related to active ingre-18 dients were predicted using the TCMSP database, and 1088 AD disease targets were obtained 19 through GeneCards, OMIM, and DisGeNET databases. By overlapping the disease targets (1088) 20 and drug targets (298), 95 key targets was identified. To further analyze these targets, functional 21 enrichment analysis (GO and KEGG) was conducted, and created a network diagram was created 22 by Cytoscape software, which highlighted the relationship between entry/pathway and key genes. 23 Furthermore, developed a PPI network for key targets and identified HUB genes (ALB, ACTB, 24 GAPDH, CASP3, and CAT), through a regulatory network of drugs-active ingredients-key targets. 25 Based on the results of network pharmacology and laboratory screening, the active ingredient to be 26 studied was selected as 6-Gingerol. Based on the results of network pharmacology and laboratory 27 screening, 6-gingerol was selected as the active ingredient to be studied, and in-vitro experiments 28 were conducted on PC12 cells treated with Aβ1-42. Select the top three proteins with the strongest 29 binding affinity (ACHE, MMP2, and PTGS2) and conduct molecular dynamics studies together with 30 CASP3 protein as the HUB gene.” is written and is not clear so revise the sentences with better clarity and understanding . also, the results are written quite short so add some numerical results and particularly add study stakeholders for which findings are useful at the end of abstract
- Introduction is written good and is covering the prior knowledge gap for the study
- What was the reason and criteria for selection of ginger as agricultural product for study is lacking so add the relevant information as criteria
- In Material and method; merge the heading “2.4 & 2.5” as it looks same
- In material and methods; “Molecular dynamics (MD) simulation analysis” dose it explains statistical analysis or not?. If not then add statistical analysis otherwise justify ?
- The headings such as “MD Simulation of CASP3-6-Gingerol Complex” contain a lot of abbreviations so try to reduce the abbreviations use in headings
- Result section is written good however check the language for grammatical mistake, consistency of findings as well as add latest references to support findings
- Do not add abbreviations in the headings and try to elaborate headings as much as possible
- Conclusion needs better elaboration along with addition of numerical results as well as lack conclusion. As “In the present investigation, active substances in ginger that regulate AD were iden-468 tified by combining network pharmacology, molecular docking, and molecular dynamics 469 simulation techniques, and the relevant mechanisms were explored. Initially, we utilized 470 the TCMSP database to predict 298 targets associated with the active ingredients” does not make impact here as it could be part of results or abstract so come up with pragmatic and concrete conclusions of study
- Grammatical mistakes observed at several places so there is need to go through the paper for language and grammatical mistakes check
Author Response
Dear reviewer,
Thank you for your comments concerning our manuscript entitled “Screening of active substances regulating Alzheimer's disease in ginger and visualization of the effectiveness on 6-Gingerol pathway targets” (Submission ID: foods-2855269). The comments and suggestions were highly insightful and enabled us to greatly improve the quality of our manuscript. We have studied comments carefully and have made correction which we hope meet with approval. Revisions in the text are shown using red color for additions or changes. The main corrections in the paper and the point-by-point response to reviewers’ comments are as following. Thank you for your suggestions. We have modified the manuscript based on your valuable modifications point by point in revised manuscript.
Major Comments
(1) The paper titled “Screening of active substances regulating Alzheimer's disease in ginger and visualization of the effectiveness on 6-Gingerol 3 pathway targets” described the role of practical application of 6-Gingero using molecular studies for the Alzheimer's disease using a series of procedure for target compounds related to active ingredients and prediction using TCMSP database along with disease obtained through GeneCards, OMIM, and DisGeNET databases. The manuscript has potential but need following changes before consideration
Response: We are grateful for your insightful and constructive comments. We carefully addressed your concerns point by point.
(2) Title is good and elaborative.
Response: We appreciate your kind words.
(3) Abstract is written poor as the methodology “In this study, 298 targets related to active ingre-18 dients were predicted using the TCMSP database, and 1088 AD disease targets were obtained 19 through GeneCards, OMIM, and DisGeNET databases. By overlapping the disease targets (1088) 20 and drug targets (298), 95 key targets was identified. To further analyze these targets, functional 21 enrichment analysis (GO and KEGG) was conducted, and created a network diagram was created 22 by Cytoscape software, which highlighted the relationship between entry/pathway and key genes. 23 Furthermore, developed a PPI network for key targets and identified HUB genes (ALB, ACTB, 24 GAPDH, CASP3, and CAT), through a regulatory network of drugs-active ingredients-key targets. 25 Based on the results of network pharmacology and laboratory screening, the active ingredient to be 26 studied was selected as 6-Gingerol. Based on the results of network pharmacology and laboratory 27 screening, 6-Gingerol was selected as the active ingredient to be studied, and in-vitro experiments 28 were conducted on PC12 cells treated with Aβ1-42. Select the top three proteins with the strongest 29 binding affinity (ACHE, MMP2, and PTGS2) and conduct molecular dynamics studies together with 30 CASP3 protein as the HUB gene.” is written and is not clear so revise the sentences with better clarity and understanding . also, the results are written quite short so add some numerical results and particularly add study stakeholders for which findings are useful at the end of abstract
Response: Thank you for your valuable suggestions. As you said, our abstract section is poorly written and indeed needs to be carefully revised. Therefore, our abstract section has been revised as follows:
Abstract: Ginger has been reported to potentially treat Alzheimer's disease (AD), but the specific compounds responsible for this biological function and their mechanisms are still unknown. In this study, a combination of network pharmacology, molecular docking, and dynamic simulation technology was used to screen active substances that regulate AD and explore their mechanisms. The TCMSP, GeneCards, OMIM, and DisGeNET databases were utilized to obtain 95 cross-targets related to ginger's active ingredients and AD as key targets. Functional enrichment analysis revealed that the pathways in which ginger's active substances may be involved in regulating AD include response to exogenous stimuli, response to oxidative stress, response to toxic substances, and lipid metabolism, among others. Furthermore, a drug-active ingredient-key target interaction network diagram was constructed, highlighting that 6-Gingerol is associated with 16 key targets. Additionally, a protein-protein interaction (PPI) network was mapped for the key targets, and HUB genes (ALB, ACTB, GAPDH, CASP3, and CAT) were identified. Based on the results of network pharmacology and cell experiments, 6-Gingerol was selected as the active ingredient for further investigation. Molecular docking was performed between 6-Gingerol and its 16 key targets, and the top three proteins with the strongest binding affinities (ACHE, MMP2, and PTGS2) were chosen for molecular dynamics analysis together with the CASP3 protein as the HUB gene. The findings indicate that 6-Gingerol exhibits strong binding ability to these disease targets, suggesting its potential role in regulating AD at the molecular level, as well as in abnormal cholinesterase metabolism and cell apoptosis, among other related regulatory pathways. These results provide a solid theoretical foundation for future in vitro experiments using actual cells and animal experiments to further investigate the application of 6-Gingerol.
Detailed Comments
(1) Introduction is written good and is covering the prior knowledge gap for the study.
Response: Thank you! Your positive words will encourage us to continue working in these field.
(2) What was the reason and criteria for selection of ginger as agricultural product for study is lacking so add the relevant information as criteria.
Response: Thanks for this point. As you suggested, we have added relevant content on the high efficacy of ginger in the introduction section. It is the reason and criteria for selection of ginger as agricultural product for study.
Ginger (Zingiber officinale, Zingiberaceae, Zingiberaceae) is a rhizome plant with both medicinal and culinary uses [24]. It is widely distributed in tropical, tropical, and temperate regions, with an annual global production of 20 million tons [25]. The easy availability of ginger contributes to its high economic value as both an agricultural product and medicine. In traditional Eastern medicine, ginger is recognized for its ability to alleviate headaches, nausea, and colds [26]. Modern medicine has also utilized ginger in the treatment of various diseases such as rheumatoid arthritis [27], atherosclerosis, ulcers [28], and depression [29]. Researchers have developed nanomaterials containing ginger's active ingredients to enhance their effects [30, 31]. Among these substances, gingerol is particularly known for its anti-inflammatory and antioxidant properties [32]. Some studies suggest that the active ingredients present in ginger may have the potential to alleviate Aβ-induced neurotoxicity by modulating oxidative stress and inflammatory pathways [33]. However, the research on the specific active compounds in ginger for the treatment of AD is still not sufficiently comprehensive. There have been no reports on identifying the compounds in ginger that are most likely to play a role in AD, and the activated pathways through which these compounds can modulate AD remain unknown.
(3) In Material and method; merge the heading “2.4 & 2.5” as it looks same.
Response: Thank you for your insightful suggestion. We have merged these two parts. As follow:
2.4. Cell experiments
PC12 cells were cultured in RPMI1640 medium (Procell Life Science & Technology Co., Ltd. ,Wuhan, China) supplemented with fetal bovine serum (FBS, 10%, v/v, obtained from Thermo Fisher, USA), penicillin (100 U/mL, obtained from Thermo Fisher, USA) and streptomycin (100 μg/mL, obtained from Thermo Fisher, USA). The cells were incubated in a humidified atmosphere of 5% CO2/95% air at 37 °C. To maintain the cell culture, PC12 cells were maintained up to 80% confluence in culture flask and passaged through digestion. Cells used for later experiment were passaged 1~2 times after resuscitation and in logarithmic growth phase, and smaller than 20 in this study.
Cell viability was evaluated by using Cell Counting Kit-8 (CCK-8, Dojindo, Japan). Before exposured, PC12 cells were seeded into 96-well plates (8×104 cells/mL) for 24 hours. Then, the culture medium was removed and replaced by the medium containing imidacloprid and acetamiprid. The cells exposed to different concentrations of Aβ1-42 (Biosynthesis Biotechnology CO.Ltd, Beijing, China) and 6-Gingerol (Acmec Biochemical Co., Ltd, Shanghai, China) was as the test group, and the control group was treated with 0.1% methanol. The test results were obtained from microplate reader (Infinite M200 PRO, TECAN, USA).
(4) In material and methods; “Molecular dynamics (MD) simulation analysis” dose it explains statistical analysis or not?. If not then add statistical analysis otherwise justify ?
Response: Thanks for this point. This section does not require statistical analysis. Molecular dynamics (MD) simulation is a computer simulation experimental method and a powerful tool for studying condensed matter systems. This technology can not only obtain the movement trajectories of atoms, but also observe various microscopic details during the movement of atoms. It is a powerful supplement to theoretical calculations and experiments. It is widely used in materials science, biophysics, drug design, etc. This study also referred to a large number of international literature during the process. None of these reports involved statistical analysis in the molecular dynamics simulation part. The following are some literature related to molecular dynamics simulation.
- Luo, Lianxiang et al. “Structure-Based Pharmacophore Modeling, Virtual Screening, Molecular Docking, ADMET, and Molecular Dynamics (MD) Simulation of Potential Inhibitors of PD-L1 from the Library of Marine Natural Products.” Marine drugs vol. 20,1 29. 25 Dec. 2021, doi:10.3390/md20010029.
- Liu, Jiaqin et al. “Dissecting the molecular mechanism of cepharanthine against COVID-19, based on a network pharmacology strategy combined with RNA-sequencing analysis, molecular docking, and molecular dynamics simulation.” Computers in biology and medicine vol. 151,Pt A (2022): 106298.
(5)The headings such as “MD Simulation of CASP3-6-Gingerol Complex” contain a lot of abbreviations so try to reduce the abbreviations use in headings
Response: Your suggestion is crucial. We have rechecked the entire text and made modifications to the abbreviations in the headings. As follow:
2.1. Screening of active ingredients in ginger and targets for Alzheimer's disease
2.2. Functional enrichment analysis of Alzheimer's disease targets
2.6. Molecular dynamics simulation analysis
(6) Result section is written good however check the language for grammatical mistake, consistency of findings as well as add latest references to support findings.
Response: Thanks for this comment. We have thoroughly checked this section for language grammar errors and ensured the consistency of the results. In addition, add some latest reference materials to support the results. There are some modifications involved, and they have been made in the manuscript.
(7) Do not add abbreviations in the headings and try to elaborate headings as much as possible
Response: Thanks for this suggestion. We have rechecked the entire text and made modifications to the abbreviations in the headings. As follow:
3.1. Active ingredients in ginger and the targets related to Alzheimer's disease
3.7. Molecular dynamics simulation analysis
3.7.1 Molecular dynamics simulation of ACHE-6-Gingerol complex
3.7.2 Molecular dynamics simulation of CASP3-6-Gingerol complex
3.7.3 Molecular dynamics simulation of MMP2-6-Gingerol complex
3.7.4 Molecular dynamics simulation of PTGS2-6-Gingerol complex
(8) Conclusion needs better elaboration along with addition of numerical results as well as lack conclusion. As “In the present investigation, active substances in ginger that regulate AD were iden-468 tified by combining network pharmacology, molecular docking, and molecular dynamics 469 simulation techniques, and the relevant mechanisms were explored. Initially, we utilized 470 the TCMSP database to predict 298 targets associated with the active ingredients” does not make impact here as it could be part of results or abstract so come up with pragmatic and concrete conclusions of study.
Response: Thank you for your comment, which helps to enhance the summary and conclusion of this work. More importantly, it can bring the important significance of the work to life on paper. As you suggested, we have made the following modifications to the text in the conclusion section:
This study combines network pharmacology, molecular docking, and molecular dynamics simulation technology to screen out the most likely active substance, 6-Gingerol, in regulating AD from the various compounds in ginger. It aims to predict, verify, and understand the related mechanisms of 6-Gingerol. Initially, the TCMSP database was used to predict 298 targets related to 143 active ingredients in ginger. Subsequently, by merging data from GeneCards, OMIM, and DisGeNET databases, 1088 AD-related targets were obtained. The overlap between disease targets (1088) and drug targets (298) resulted in 95 cross-targets, which were defined as critical targets. Functional enrichment analysis, including GO annotation and KEGG pathway analysis, was performed on these 95 targets. The pathways in which the active substances in ginger may be involved in regulating AD include response to exogenous stimuli, response to oxidative stress, toxic substance reactions, lipids and atherosclerosis, diabetic cardiomyopathy, etc. Furthermore, the drug - active ingredients - key targets interaction network diagram was constructed using Cytoscape. This diagram showcases the relationship between active ingredients and key targets, providing a more intuitive representation of ginger's role in regulating AD and its associated metabolic pathways. The findings of this study serve as a basis for screening effective active substances in ginger. Additonaly, PPI networks were developed to identify HUB genes (ALB, ACTB, GAPDH, CASP3, and CAT) through careful screening. Based on the results of network pharmacology and laboratory screening, 6-Gingerol was selected as the active ingredient for studying its effects on PC12 cells. The molecular dynamics studies focused on the top three proteins (ACHE, MMP2, and PTGS2) with the strongest binding affinity, along with the CASP3 protein as the HUB gene. The results indicate that 6-Gingerol exhibits strong binding ability to these disease targets, suggesting its potential role in regulating Alzheimer's disease at the molecular level and cholinesterase metabolism abnormalities, cell apoptosis, and other related regulatory pathways. These findings provide a solid theoretical foundation for future in-vitro and animal experiments involving the application of 6-Gingerol.
Comments on the Quality of English Language
Grammatical mistakes observed at several places so there is need to go through the paper for language and grammatical mistakes check
Response: Thank you very much. We have revised the language and grammar sections of the article according to your suggestions. There are some modifications involved, and they have been made in the manuscript.
Thanks a lot for your comments again. We truly appreciate it that enhances our work to a great extent. Without your help, our conclusion is weak.
Sincerely,
Author
